# Planning with a Learned Policy Basis to Optimally Solve Complex Tasks

**Primary Keywords:** *(2) Learning*

## Abstract

Conventional reinforcement learning (RL) methods can successfully solve a wide range of sequential decision problems. However, learning policies that can generalize predictably across multiple tasks in a setting with non-Markovian reward specifications is a challenging problem. We propose to use successor features to learn a set of local policies that each solves a well-defined subproblem. In a task described by a finite state automaton (FSA) that involves the same set of subproblems, the combination of these local policies can then be used to generate an optimal solution without additional learning. In contrast to other methods that combine local policies via planning, our method asymptotically attains global optimality, even in stochastic environments.

## Introduction

Autonomous agents that interact with an environment usually face tasks that comprise complex, entangled behaviors over long horizons. Conventional RL methods have successfully addressed this. However, in cases when the agent is meant to perform several tasks across similar environments, training a policy for every task separately can be time-consuming and requires a lot of data. In such cases, the agent can utilize a method that has built-in generalization capabilities. One such method relies on the assumption that reward functions of these tasks can be decomposed into a linear combination of successor features (Barreto et al. 2017). When a new task is presented, it is possible to combine previously learned policies and their successor features to solve a new task. While the combination of such policies is guaranteed to be better than any previously learned policy, it need not be optimal. However, as was recently shown by Alegre, Bazzan, and Da Silva (2022), one can utilize recent advancement in multiobjective RL to learn a set of policies that contains an optimal policy for any linear combination of successor features.

While these and other conventional RL methods use Markovian reward functions, expressing a task with such reward function can be difficult and may not even be possible in some cases (Whitehead and Lin 1995). In settings where the reward function cannot be expressed in Markovian terms, task specification has raised especial interest in the last few years (Toro Icarte et al. 2018a; Camacho et al. 2019). In our work we focus on developing a method that utilizes generalization capabilities of successor features in a settings with non-Markovian reward functions.

Prior techniques for such settings have been proposed in contexts where the existence of a set of propositional symbols allowed for the definition of high-level tasks by the means of logics (Vaezipoor et al. 2021; Toro Icarte et al. 2019) or finite state automata (Toro Icarte et al. 2018a). Similar to several hierarchical reinforcement learning methods, they are often based on decomposing tasks into sub-tasks and solving each sub-task independently (Dietterich 2000; Sutton, Precup, and Singh 1999). While this has an advantage that solving parts of a task in isolation is often simpler, the disadvantage of such approach is that the combination of optimal solutions to sub-tasks can lead to a sub-optimal overall policy. This is commonly referred to as recursive optimality (Dietterich 2000) or myopic policy (Vaezipoor et al. 2021).

To alleviate this issue, one can consider methods that condition the policy or the value function on the specification of the whole task (Schaul et al. 2015) and such approaches were recently also proposed for tasks with non-Markovian reward functions (Vaezipoor et al. 2021). However, the methods that specify the whole task usually rely on a blackbox neural network (NN) for planning when determining which sub-goal to reach next. This makes it hard to interpret the plan to solve the task and although they achieve promising empirical results it is unclear whether and when these approaches will generalize to a new task.

Instead, our work aims to use task decomposition without sacrificing global optimality to achieve predictable generalization. The method we propose learns a set of *local* policies in sub-tasks such that their combination forms a *globally optimal* policy for a large collection of problems described with FSAs. A new policy that solves any new task can then be created without additional learning by planning on a given FSA task description. Our contributions are:

- We propose to use successor features to learn a policy basis that is suitable for planning in stochastic domains.

- We develop a planning framework that uses such policy bases for zero-shot generalization to complex temporal tasks described by an arbitrary FSA.

- We prove that if the policies in this basis are optimal, our framework produces a globally optimal solution even in stochastic domains.

## Background and notation

Given a finite set $\mathcal{X}$, let $\Delta(\mathcal{X}) = \{p \in \mathbb{R}^{\mathcal{X}} : \sum_x p(x) = 1, p(x) \geq 0 \ (\forall x)\}$ denote the probability simplex on $\mathcal{X}$. Given a probability distribution $q \in \Delta(\mathcal{X})$, let $\mathrm{supp}(q) = \{x \in \mathcal{X} : q(x) > 0\} \subseteq \mathcal{X}$ denote the support of $q$.

### Reinforcement Learning

Reinforcement learning problems commonly assume an underlying Markov Decision Process (MDP). We define an MDP as the tuple $\mathcal{M} = \langle \mathcal{S}, \mathcal{E}, \mathcal{A}, \mathcal{R}, \mathbb{P}_0, \mathbb{P}, \gamma \rangle$ where $\mathcal{S}$ is the set of states, $\mathcal{E}$ is the set of exit states, $\mathcal{A}$ is the action space, $\mathcal{R} : \mathcal{S} \times \mathcal{A} \times \mathcal{S} \to \mathbb{R}$ is the reward function, $\mathbb{P}_0 \in \Delta(\mathcal{S})$ is the probability distribution of initial states, $\mathbb{P} : \mathcal{S} \times \mathcal{A} \to \Delta(\mathcal{S})$ is the transition probability function and $0 \leq \gamma < 1$ is the discount factor. The set of exit states $\mathcal{E}$ induces a set of terminal transitions $T = (\mathcal{S} \setminus \mathcal{E}) \times \mathcal{A} \times \mathcal{E}$.

The learning agent interacts in an episodic manner with the environment following a policy $\pi : \mathcal{S} \to \mathcal{A}$. At each timestep, the agent observes a state $s_t$, chooses the action $a_t = \pi(s_t)$, transitions to a new state $s_{t+1} \sim \mathbb{P}(\cdot | s_t, a_t)$ and receives a reward $\mathcal{R}(s_t, a_t, s_{t+1})$. The episode ends when the agent observes a terminal transition $(s_t, a_t, s_{t+1}) \in T$ and a new episode starts with initial state $s_0 \sim \mathbb{P}_0(\cdot)$.

The goal of the agent is to find an optimal policy $\pi^*$ that maximizes the expected discounted return, for any state-action pair $(s, a) \in \mathcal{S} \times \mathcal{A}$,

$$Q^{\pi}(s, a) = \mathbb{E}_{\pi}\left[\sum_{i=t}^{\infty} \gamma^{i-t} \mathcal{R}_i \,\middle|\, S_t = s, A_t = a\right], \quad (1)$$

where $\mathcal{R}_i = \mathcal{R}(S_i, A_i, S_{i+1})$. Hence, an optimal policy is

$$\pi^* \in \arg\max_{\pi} Q^{\pi}(s, a) \ \ \forall (s, a) \in \mathcal{S} \times \mathcal{A}$$

with ties broken arbitrarily. The action value function defined in Equation (1) satisfies the recursive Bellman equation

$$Q^{\pi}(s, a) = \mathbb{E}_{s' \sim \mathbb{P}(\cdot | s, a)}\left[\mathcal{R}(s, a, s') + V^{\pi}(s')\right] \quad (2)$$

for any $(s, a) \in \mathcal{S} \times \mathcal{A}$. The state value function is obtained by maximizing the action value function over the actions, $V^{\pi}(s) = \max_a Q^{\pi}(s, a) \ \forall s \in \mathcal{S}$. Throughout the paper we use $Q^*$ and $V^*$ to refer to the optimal, respectively, action and state value functions.

### Successor Features

Successor features (SFs) (Dayan 1993; Barreto et al. 2017) is a widely used RL representation framework that assumes the reward function is linearly expressible with respect to a feature vector,

$$\mathcal{R}^{\mathbf{w}}(s, a, s') = \mathbf{w}^{\intercal} \phi(s, a, s'). \quad (3)$$

Here, $\phi : \mathcal{S} \times \mathcal{A} \times \mathcal{S} \to \mathbb{R}^d$ maps transitions to feature vectors and $\mathbf{w} \in \mathbb{R}^d$ is a weight vector. Every weight vector $\mathbf{w}$ induces a different reward function and, thus, a task. The SF vector of a state-action pair $(s, a) \in \mathcal{S} \times \mathcal{A}$ under a policy $\pi$ is the expected discounted sum of feature vectors following ,

$$\psi^{\pi}(s, a) = \mathbb{E}_{\pi}\left[\sum_{i=t}^{\infty} \gamma^{i-t} \phi_i \,\middle|\, S_t = s, A_t = a\right], \quad (4)$$

where $\phi_i = \phi(S_i, A_i, S_{i+1})$. The action value function for a state-action pair $(s, a)$ under policy $\pi$ can be efficiently represented using the SF vector. Due to the linearity of the reward function, the weight vector can be decoupled from the Bellman recursion. Following the definition of Equations (1) and (3), the action value function in the SF framework can be rewritten as

$$\begin{aligned} Q^{\pi}_{\mathbf{w}}(s, a) &= \mathbb{E}_{\pi}\left[\sum_{i=t}^{\infty} \gamma^{i-t} \mathbf{w}^{\intercal} \phi_i \,\middle|\, S_t = s, A_t = a\right] \\ &= \mathbf{w}^{\intercal} \mathbb{E}_{\pi}\left[\sum_{i=t}^{\infty} \gamma^{i-t} \phi_i \,\middle|\, S_t = s, A_t = a\right]. \quad (5) \end{aligned}$$

Once the SF representation is obtained, this allows *generalized policy evaluation* (GPE) as a consequence of Equation (5) (Barreto et al. 2020). Simultaneously, *generalized policy improvement* (GPI) can be used to obtain new better policies (Barreto et al. 2017).

A family of MDPs is defined as the set of MDPs that share all the components, except the reward function. This set can be properly defined as

$$\mathcal{M}^{\phi} \equiv \{\langle \mathcal{S}, \mathcal{E}, \mathcal{A}, \mathcal{R}_{\mathbf{w}}, \mathbb{P}_0, \mathbb{P}, \gamma \rangle | \mathcal{R}_{\mathbf{w}} = \mathbf{w}^{\intercal} \phi, \forall \mathbf{w} \in \mathbb{R}^d\}.$$

Transfer learning on families of MDPs is possible thanks to GPI. Given a set of policies $\Pi$, learned on the same family $\mathcal{M}^{\phi}$, for which their respective SF representations have been computed, and a new task by $\mathbf{w}' \in \mathbb{R}^d$, a GPI policy $\pi_{\mathrm{GPI}}$ is derived as

$$\pi_{\mathrm{GPI}}(s) \in \arg\max_{a \in \mathcal{A}} \max_{\pi \in \Pi} Q^{\pi}_{\mathbf{w}'}(s, a) \ \forall s \in \mathcal{S}. \quad (6)$$

The new policy $\pi_{\mathrm{GPI}}$ is known to perform no worse than any policy $\pi \in \Pi$ in the new task $\mathbf{w}'$ (Barreto et al. 2017, cf. Theorem 1). However, there is no guarantee with respect to the optimal behavior for such task. A fundamental question to solve the so-called *optimal transfer learning problem* is which policies should be included in the set of policies $\Pi$ so the optimal policy for any weight vector $\mathbf{w} \in \mathbb{R}^d$ can be derived.

### Convex Coverage Set of Policies

The recent work of Alegre, Bazzan, and Da Silva (2022) solves the optimal transfer learning problem. They draw the connection between the SF transfer learning problem and multi-objective RL (MORL). The pivotal fact is that the SF representation (4) can be interpreted as a multi-dimensional value function and then solved using multi-objective optimization techniques. They use the optimistic linear support (OLS) algorithm to learn a set of policies that constitutes a convex coverage set (CCS) (Roijers, Whiteson, and Oliehoek 2015). In terms of its multi-objective value, $V^{\pi}_{\mathbf{w}} = \mathbb{E}_{S_0 \sim \mathbb{P}_0}[V^{\pi}_{\mathbf{w}}(S_0)]$, set $\Pi_{\mathrm{CCS}}$ contains all non-dominated policies,

$$\begin{aligned} \Pi_{\mathrm{CCS}} &= \{\pi \mid \exists \mathbf{w} \text{ s.t. } \forall \psi^{\pi'}, \ \mathbf{w}^{\intercal} \psi^{\pi} \geq \mathbf{w}^{\intercal} \psi^{\pi'}\} \\ &= \{\pi \mid \exists \mathbf{w} \text{ s.t. } \forall \pi', \ V^{\pi}_{\mathbf{w}} \geq V^{\pi'}_{\mathbf{w}}\}. \quad (7) \end{aligned}$$

Intuitively, all policies in $\Pi_{\mathrm{CCS}}$ are optimal in at least one task $\mathbf{w} \in \mathbb{R}^d$. This also means that for every task $\mathbf{w}^d$ there exists $\pi \in \Pi_{\mathrm{CCS}}$ such that $\pi$ is optimal for $\mathbf{w}$.

## Propositional Logic

We assume that environments are endowed with set of high-level, boolean-valued propositional symbols $\mathcal{P}$ and that they are associated with the set of exit states $\mathcal{E}$ of a low-level MDP $\mathcal{M}$. Every transition $\in \mathcal{S} \times \mathcal{A} \times \mathcal{S}$ induces some propositional valuation (assignment of truth values) $2^{\mathcal{P}}$. Such a valuation depends on the new state and occurs under a mapping $\mathcal{O} : \mathcal{S} \rightarrow 2^{\mathcal{P}}$ that is known to the agent. Nonetheless, only exit states $\varepsilon \in \mathcal{E}$ make propositions true under $\mathcal{O}$. We assume that that propositional symbols are mutually exclusive, and the agent cannot observe two symbols in the same transition. We say that a valuation $\Gamma$ satisfies a propositional symbol $p$, formally $\Gamma \vDash p$, if $p$ is true in $\Gamma$. E.g., in the office domain depicted in Figure 1a, the propositional symbols are $\mathcal{P} = \{\text{🐾}, \boxtimes, o\}$ while the exit states $\mathcal{E} = \{\text{🐾}^1, \text{🐾}^2, \boxtimes^1, \boxtimes^2, o^1, o^2\}$. Consequently, $\mathcal{O}(\text{🐾}^1) \vDash \text{🐾}$ and $\mathcal{O}(\text{🐾}^2) \vDash \text{🐾}$.

## Finite State Automaton

Task instructions can be specified via a finite state automaton. These are tuples $\mathcal{F} = \langle \mathcal{U}, u_0, \mathcal{T}, L, \delta \rangle$ where $\mathcal{U}$ is the finite set of states, $u_0 \in \mathcal{U}$ is the initial state, $\mathcal{T}$ is the set of terminal states with $\mathcal{U} \cap \mathcal{T} = \emptyset$, $L : \mathcal{U} \times (\mathcal{U} \cup \mathcal{T}) \rightarrow 2^{\mathcal{P}}$ is a labelling function that maps FSA states transitions to truth values for the propositions and $\delta : \mathcal{U} \rightarrow \{0, 1\}$ is a high-level reward function. Each transition among FSA states $(u, u')$ defines a subgoal. The agent has to observe some propositional valuation $L(u, u')$ in order to achieve it and FSA states can only be connected by a subgoal. Non-existing transitions $(u, u')$ get mapped to $L(u, u') = \bot$. In Figure 4c, the FSA state $u_0$ has two outgoing subgoals: getting mail (labeled as $\boxtimes$) and getting coffee (labeled as 🐾). The reward function $\delta$ gives a reward larger than 0 only to terminal states. In other words, such a reward function is $\delta(u) = 0 \ \forall u \in \mathcal{U}$ and $\delta(\mathbf{t}) = 1 \ \forall \mathbf{t} \in \mathcal{T}$.

# Using Successor Features to Solve non-Markovian Reward Specifications

We focus on the setting in which a low-level MDP is equipped with a reward structure like in Equation (3). We let the low-level be represented by a family of MDPs $\mathcal{M}^{\phi}$, where each weight vector $\mathbf{w} \in \mathbb{R}^d$ specifies a low-level task. The agent receives high-level task specifications in the more flexible form of an FSA which permits the specification of non-Markovian reward structures. The combination of a low-level family of MDPs and a high-level FSA gives rise to a product MDP $\mathcal{M}' = \mathcal{F} \times \mathcal{M}^{\phi}$, that satisfies the Markov property and where the state space is augmented to be $\mathcal{U} \times \mathcal{S}$.

A product MDP $\mathcal{M}'$ is a well-defined MDP and can be solved with conventional RL methods such as Q-learning (Watkins and Dayan 1992). This is, however, impractical since policies should be retrained every time a new high-level task is specified. Exploiting the problem structure is essential for tractable learning, where components can be reused for new task specifications. The special reward structure of the low-level MDPs and our particular choice of feature vectors, later introduced, allow us to define an algorithm able to achieve a solution by simply planning in the space of augmented exit states $\mathcal{U} \times \mathcal{E}$. This inherently makes obtaining the optimal policy more efficient than solving the whole product MDP, as we reduce the number of states on which it is necessary to compute the value function.

When presented with different task specifications (e.g. Figure 4), the agent may have to perform the same subtask at different moments of the plan or in different FSAs. We aim to provide agents with a collection of base behaviors that can be combined to retrieve the optimal behavior.

In line with the previous reasoning, we introduce a two-step algorithm in which the agent first learns a $\Pi_{\text{CCS}}$ (a set of policies that constitute a CCS) on a well-specified representation of the environment. Then these subpolicies are used to solve efficiently any FSA task specification on the propositional symbols of the environment. In what follows, we motivate the design of the feature vectors, explain our high-level dynamic programming algorithm and prove that it achieves the optimal solution.

**Feature vectors:** For a family of MDPs $\mathcal{M}^{\phi}$, feature vectors $\phi(s, a, s')$ are $|\mathcal{E}|$-dimensional. Each feature component $\phi_j$ is associated with an exit state $\varepsilon_j \in \mathcal{E} = \{\varepsilon_1, \ldots, \varepsilon_{|\mathcal{E}|}\}$. Such vectors are built as follows. At terminal transitions $(s, a, \varepsilon_i) \in T$, $\phi_j = 1$ when $j = i$ and $\phi_j = 0$ when $j \neq i$. For non-terminal transitions, we just require that $\mathbf{w}^{\intercal}\phi(s, a, s') < 1$. In the case that $\phi(s, a, s') = \mathbf{0} \in \mathbb{R}^{|\mathcal{E}|}$ for non-terminal transitions, the SF representation in Equation (4) of each policy consists of a discounted distribution over the exit states. This indicates how likely it is to reach each exit state following such a policy. Furthermore, we require that $\mathcal{E} \subset \text{supp}(\mathbb{P}_0)$ so the value functions at exit states are well-defined.

---

**Algorithm 1: SF-FSA-VI**

---

**Input:** Low-level MDP $\mathcal{M}^{\phi}$, task specification $\mathcal{F}$
1: Obtain $\Pi_{\text{CCS}}$ on $\mathcal{M}^{\phi}$.
2: Initially $\mathbf{w}^0(u) = \mathbf{0} \in \mathbb{R}^{|\mathcal{E}|} \ \ \forall u \in \mathcal{U}$.
3: **while** not done **do**
4:     **for** $u \in \mathcal{U}$ **do**
5:         Update each $\mathbf{w}_j^{k+1}(u)$ with Equation (10).
6: **return** $\{\mathbf{w}^*(u) \ \forall u \in \mathcal{U}\}$

---

**Algorithm** The solution to an FSA task specification implies solving a product MDP $\mathcal{M}' = \mathcal{F} \times \mathcal{M}^{\phi}$. Since we have the CCS, the optimal Q-function can be represented by a weight vector $\mathbf{w}^*$:

$$Q^*(u, s, a) = \max_{\pi \in \Pi_{\text{CCS}}} \mathbf{w}^*(u)^{\intercal}\psi^{\pi}(s, a). \tag{8}$$

for all $(u, s, a) \in \mathcal{U} \times \mathcal{S} \times \mathcal{A}$. Here, $\mathbf{w}_j^*(u)$ indicates the optimal value from exit state $\varepsilon_j \in \mathcal{E}$ for FSA state $u$. Thus, we observe that finding the optimal weight vectors $\mathbf{w}^*(u)$, $\forall u \in \mathcal{U}$, is enough for retrieving the optimal action value function of the product MDP $\mathcal{M}'$. We can obtain this vector using a dynamic-programming approach similar to value

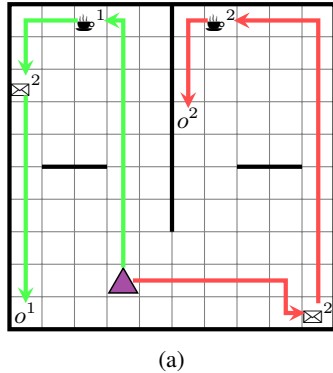



| (a) | (b) |

Figure 1: Depiction of the Office (a) and Delivery (b) environments, FSA task specification of the composite task in the Office domain and the FSA task specification of the sequential task in the Delivery domain (b). In (a) $\mathcal{P} = \{\text{☕}, \boxtimes, o\}$ and $\mathcal{E} = \{\text{☕}^1, \text{☕}^2, \boxtimes^1, \boxtimes^2, o^1, o^2\}$. In (b), $\mathcal{E} = \mathcal{P} = \{A, B, C, H\}$.

iteration:

$$\mathbf{w}_j^{k+1}(u) = \max_a Q^*(u, \varepsilon_j, a) \tag{9}$$

$$= \max_{a,\pi} \mathbf{w}^k \big(\tau(u, \mathcal{O}(\varepsilon_j))\big)^\intercal \boldsymbol{\psi}^\pi(\varepsilon_j, a), \tag{10}$$

where $\tau(u, \mathcal{O}(\varepsilon)) \in \mathcal{U}$ is the FSA state that results from achieving the valuation $\mathcal{O}(\varepsilon)$ in $u$. We know that, $\mathbf{w}_j^k(u) = 1$ if $\tau(u, \mathcal{O}(j)) = \mathbf{t}$, per definition, since the high-level reward function $\delta(\mathbf{t}) = 1$. As $k \to \infty$, our algorithm converges to the optimal set of weight vectors $\{\mathbf{w}^*(u)\}_{u \in \mathcal{U}}$ and, as a result, to the optimal value function in Equation (8).

**Proof of optimality** We first restate the following theorem from Alegre, Bazzan, and Da Silva (2022).

**Theorem 1** (Alegre, Bazzan, and Silva, 2022)**.** *Let $\Pi$ be a set of policies such that the set of their expected SFs, $\Psi = \{\boldsymbol{\psi}^\pi\}_{\pi \in \Pi}$, constitutes a CCS. Then, given any weight vector $\mathbf{w} \in \mathbb{R}^d$, the GPI policy $\pi_{\mathbf{w}}^{GPI}(s) \in \arg\max_{a \in A} \max_{\pi \in \Pi} Q_{\mathbf{w}}^\pi(s, a)$ is optimal with respect to $\mathbf{w} : V_{\mathbf{w}}^{GPI} = V_{\mathbf{w}}^*$.*

Applied to our setting, once the set of policies $\Pi_{\text{CCS}}$ and associated SFs have been computed, we can define an arbitrary vector $\mathbf{w}$ of rewards on the exit states, and use the CCS to obtain an optimal policy $\pi_{\mathbf{w}}^*$ and an optimal value function $V_{\mathbf{w}}^*$ without learning. We can then use composition by setting the reward of the exit states equal to the optimal value.

We aim to show that for each augmented state $(u, s) \in \mathcal{U} \times \mathcal{S}$, the value function output by our algorithm equals the optimal value of $(u, s)$ in the product MDP $\mathcal{M}' = \mathcal{F} \times \mathcal{M}^\phi$, i.e. that $V_{\mathbf{w}(u)}(s) = V_{\mathcal{M}'}^*(u, s)$. To do so, it is sufficient to show that the weight vectors $\{\mathbf{w}(u)\}_{u \in \mathcal{U}}$ are optimal.

Each element of $\mathbf{w}(u)$ is recursively defined as $\mathbf{w}_j(u) = V_{\mathbf{w}(\tau(u, \mathcal{O}(\varepsilon_j)))}(\varepsilon_j)$. If all weight vectors are optimal, it holds that $V_{\mathbf{w}(\tau(u, \mathcal{O}(\varepsilon_j)))}(\varepsilon_j) = V_{\mathcal{M}'}^*(\mathbf{w}(\tau(u, \mathcal{O}(\varepsilon_j))), \varepsilon_j)$ for each such exit state. Due to the above theorem, the value function $V_{\mathbf{w}(u)}$ is optimal for $\mathbf{w}(u)$. Due to composition that follows GPE and GPI, this means that the value of each internal state $s$ is optimal, i.e. that $V_{\mathbf{w}(u)}(s) = V_{\mathcal{M}'}^*(u, s)$.

It remains to show that the weight vectors $\{\mathbf{w}(u)\}_{u \in \mathcal{U}}$ returned by the algorithm are indeed optimal. To do so it is sufficient to focus on the set of augmented exit states $\mathcal{U} \times \mathcal{E}$. We can state a set of optimality equations on the weight vectors as follows:

$$\mathbf{w}_j^*(u) = V_{\mathbf{w}^*(\tau(u, \mathcal{O}(\varepsilon)))}(\varepsilon_j) = \max_a Q^*(\tau(u, \mathcal{O}(\varepsilon)), \varepsilon_j, a)$$

$$= \max_a \max_\pi \boldsymbol{\psi}^\pi(\varepsilon_j, a)^\intercal \mathbf{w}^*(\tau(u, \mathcal{O}(\varepsilon))),$$

where $\boldsymbol{\psi}^\pi(\varepsilon_j, a) = \sum_{s'} \mathbb{P}(s'|\varepsilon_j, a) \boldsymbol{\psi}^\pi(\varepsilon_j, a, s')$. Due to the discount factor $\gamma$, we have $\|\boldsymbol{\psi}(\varepsilon_j, a)\|_1 < 1$. Hence the update rule in (10) is a contraction and converges to the set of optimal weight vectors.

## Experiments

We test our method in two complex discrete environments. At test time, we change the reward to $-1$ for every timestep and use the cumulative reward as performance metric. We report two types of results. First, we are interested in how fast our method can build a global solution for the high-level task specification as it learns the base subpolicies. For this, we let the algorithm learn the base sub-subpolicies and after some interactions fully retrain the high-level policy and test its performance. Second, once the base behaviors are learned, we measure how many of iterations Algorithm 1 needs to converge to an optimal solution for different task specifications. In any case, we compare against existing baselines.

**Environments and tasks** We use the Delivery domain (Araki et al. 2021) and a modified version of the Office domain (Toro Icarte et al. 2018a) as test beds for our algorithm. Both environments are depicted in Figure 1 and present a propositional vocabulary that is rich enough to build complex tasks. In the Delivery domain there is a single low-level state associated with each of the propositional symbols, implying that $\mathcal{E} = \mathcal{P} = \{A, B, C, H\}$. The feature vectors are consistent with our design choice. For terminal transitions, they correspond to their one-hot encodings of the terminal states. There exist obstacle states (in black) for which, upon entering, the feature vector is $\phi(s, a, s') = -\mathbf{1000} \in \mathbb{R}^4$.

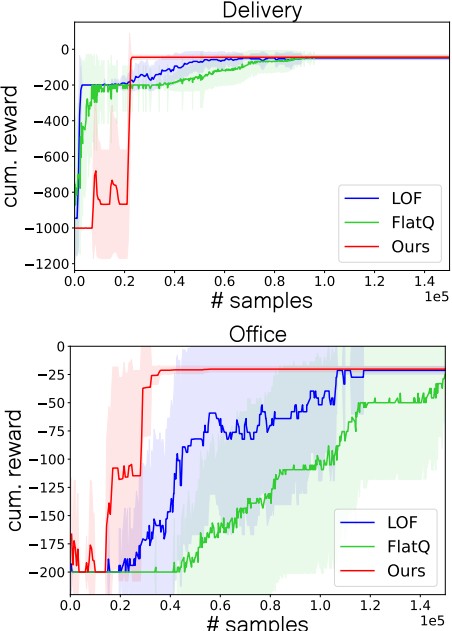
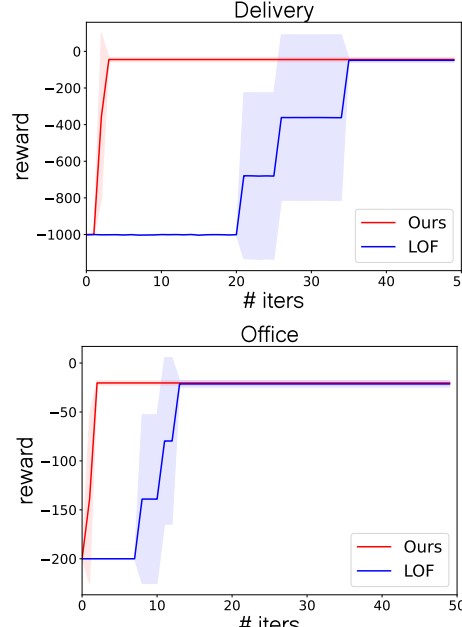

Figure 2: Experimental results for learning (Delivery, top-left and Office, bottom-left) and compositionality (Delivery, top-right and Office, bottom-right). Results show the average performance and standard deviation over the three tasks and 5 seeds per task.

This transforms in a large negative reward when multiplied with a corresponding weight vector $\mathbf{w} \in \mathbb{R}^4$. For regular grid cells (in white) $\phi(s, a, s') = \mathbf{0} \in \mathbb{R}^4$. The Office domain is more complex since there are three propositional symbols $\mathcal{P} = \{\text{☕}, \boxtimes, o\}$ which can be satisfied at different locations, namely $\mathcal{E} = \{\text{☕}^1, \text{☕}^2, \boxtimes^1, \boxtimes^1, o^1, o^2\}$. Here, there are no obstacle states and $\phi(s, a, s') = \mathbf{0} \in \mathbb{R}^6$ for non-terminal transitions.

For each of the environments we define three different tasks: sequential, disjunction and composite (all described in Figure 4). The sequential task is meant to show how our algorithm can indeed be effectively used to plan over long horizons, when the other two tasks show the ability of our method to optimally compose the base subpolicies in complex settings. In natural language, the tasks in the Delivery domain correspond to: "go to $A$, then $B$, then $C$ and finally $H$" (sequential), "go to $A$ or $B$, then $C$ and finally $H$" (disjunction) and "go to $A$ and $B$ in any order, then $B$, then $C$ and finally $H$" (composite). The agent has to complete the tasks by avoiding obstacles. The counterpart of these tasks in the Office environment are: "get a coffee, then pick up mail and then go to an office" (sequential), "get a coffee or mail, and then go to an office" (disjunction) and "get a coffee and mail in any order, and then go to an office" (composite). Our agent never learns how to solve these tasks, but rather learns the set of subpolicies that constitutes the CCS. At test time, we provide the agent with the FSA task specification, extract the high-level value function and test its performance on solving the task.

**Baselines** In the literature, we find the most similar approach to ours in the Logical Options Framework

(LOF, (Araki et al. 2021)). We thus use LOF and flat Q-learning on the product MDP as baselines. LOF trains one option per exit state, which are trained simultaneously using intra-option learning, and then uses a high-level value iteration algorithm to train a meta-policy that decides which option to execute in each of the MDP states. On the other hand, the latter learns the action value function in the flat product MDP, from which it extracts the policy. Under certain conditions, flat Q-learning converges to the optimal value function but, especially for longer tasks, it may take a large number of samples. Additionally, it is trained for a specific task, so it is not able to generalize to other task specifications. For LOF, we followed the implementation details prescribed by the authors.

## Results

Empirical results are shown in Figure 2. The plots on the left column reflect how LOF, flat Q-learning and our method perform at solving an FSA task specification during the learning phase. Results are averaged over the three tasks (sequential, disjunction and composite) previously described for each environment. In both environments, our method is the first to reach optimal performance. There exist, however, some differences between LOF and our method. LOF trains all options simultaneously with intra-option learning. This means that, every transition $(s, a, s)$ is used to update all options' value functions and policies. In our case, the learning of a $\Pi_{\text{CCS}}$ is done sequentially, with a fixed sample budget per subpolicy, we use a total of $8 \cdot 10^3$ samples per subpolicy in both environments. Both options and the SF representation of subpolicies are learned using Q-learning. Though it is true that our method uses experience replay to speed up the sub-

policies' learning, the sequential learning may be seen as a disadvantage with respect to LOF since it needs to have fully learned some base behaviors to be able to build a feasible solution. This is clearly observed in the Delivery domain (Figure 2, top left), where at the early stages of the interaction, our method achieves very low cumulative reward. This is because it takes some time for the agent to learn enough subpolicies and can offers a plausible solution. Nonetheless, it eventually converges to the optimal policy. Similarly, LOF converges to the optimal policy albeit it takes slightly longer to learn. In the more complex Office environment, results follow the same pattern. However, this environment breaks one of the of LOF requirements for optimality: to have a single exit state associated with each predicate. In this problem, for each predicate there exist two exit states that can satisfy them. This makes LOF prone to converge to suboptimal solutions when our algorithm attains optimality. This is the case for the composite task, where LOF is short-sighted and returns a longer path (in red, Figure 1a) in contrast to ours that retrieves the optimal solution (in green, Figure 1a). This means that our method is more flexible in the task specification. In this environment, our algorithm also converge faster with a more obvious gap with LOF. In any case, learning sub-policies or options is faster than learning on the flat product MDP, as flat Q-learning takes the longest to converge.

The right column in Figure 2 shows that our algorithm needs less iterations to learn an optimal representation of the global optimal value function via planning when compared to LOF. In LOF, the cost of each iteration of value iteration is $|\mathcal{U}| \times |\mathcal{S}| \times |\mathcal{K}|$, where $\mathcal{K}$ is the set of options, while for the Algorithm 1 we propose it is $|\mathcal{U}| \times |\mathcal{E}| \times |\Pi_{\text{CCS}}|$. By definition, the number of options is equivalent to the number of exit states $|\mathcal{K}| = |\mathcal{E}|$, so a single iteration of our method is more efficient than LOF whenever $|\Pi_{\text{CCS}}| \ll |\mathcal{S}|$. In our experiments, the sizes of the CCS are $15$ and $12$ for the Delivery and Office domains, respectively, while the sizes of the state spaces are of $225$ and $121$. Therefore, since our algorithm needs less shorter iterations during planning, it outperforms LOF in terms of planning speed in both domains when composing the global solution.

In deterministic environments, it is sufficient to learn the subpolicies associated with the extrema weights ( i.e. those subpolicies that reach each of the exit states individually) to find a globally optimal policy via planning. In such cases, it may not be necessary to learn a full CCS. That is why, approaches that use the options framework such as LOF traditionally define one option per subgoal. However, there are scenarios, in which these approaches will not find optimal policy. This is the case for most stochastic environments. For example, consider the very simple domain of Double Slit in and the FSA task specification in Figure 3. In this environment, there are two exit states $\mathcal{E} = \{\text{blue}, \text{red}\}$. The agent starts in the leftmost column and middle row. At every timestep, the agent chooses an action amongst $\{\text{UP}, \text{RIGHT}, \text{DOWN}\}$ and is pushed one column to the right in addition to moving in the chosen direction, except in the last column. If the agent chooses RIGHT, he moves an extra column to the right. At every timestep there is a random wind that can blow the agent away up to three positions in the vertical direction. The

FSA task specification represents a task in which the agent is indifferent between achieving either of the goal states. Since the RIGHT action brings the agent closer to both goals, the optimal behavior in this case is to commit to either goal as late as possible. In this setting, methods that use one policy per sub-goal, such as LOF, train two policies to reach both goals. This means that the agent has to commit to one of the goals from the very beginning, which hurts the performance as it has to make up for the consequences of the random noise. On the other hand, the CCS used by our method will contain an additional policy that is indifferent between two goals. This leads to a performance gap as our approach achieves an average accumulated reward of $-19.7 \pm 3.65$ and LOF $-22.70 \pm 5.72$.

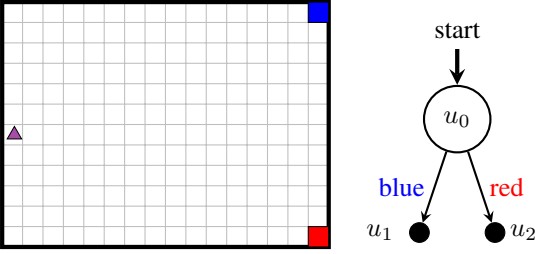

Figure 3: Double Slit environment (left) and FSA task specification to reach either goal locations blue or red (right).

## Related Work

One of the key distinctions in our research compared to prior studies is the optimality of the final solution. As noted by Dietterich (2000), hierarchical methods usually have the capability to achieve hierarchical, recursive, or global optimality. The challenge that often arises when sub-task policies are trained in isolation is that the combination of these locally optimal policies does not lead to a globally optimal policy but a recursively (Dayan and Hinton 1992) or hierarchically optimal policy (Sutton, Precup, and Singh 1999; Mann, Mannor, and Precup 2015; Araki et al. 2021). To tackle this challenge, our approach relies on acquiring a set of low-level policies for each sub-task and employing planning to identify the optimal combination of low-level policies when solving a particular task. By learning the CCS with OLS (Roijers, Whiteson, and Oliehoek 2014) in combination with high-level planning our approach ensures that globally optimal policy is found. In this regard, the work of Alegre, Bazzan, and Da Silva (2022) is of particular interest as it was the first work that used OLS and successor features (Barreto et al. 2017) for optimal transfer learning. However, this method has only applied in a setting with Markovian reward function and has not been used with non-Markovian task specifications or high-level planning.

On the other hand, many recent approaches proposed to use high-level task specifications in the form of LTL (Toro Icarte et al. 2018b; Kuo, Katz, and Barbu 2020; Vaezipoor et al. 2021; Jothimurugan et al. 2021), or similar formal language specifications (Toro Icarte et al. 2019; Camacho et al. 2019; Araki et al. 2021; Toro Icarte et al. 2022) to learn policies.

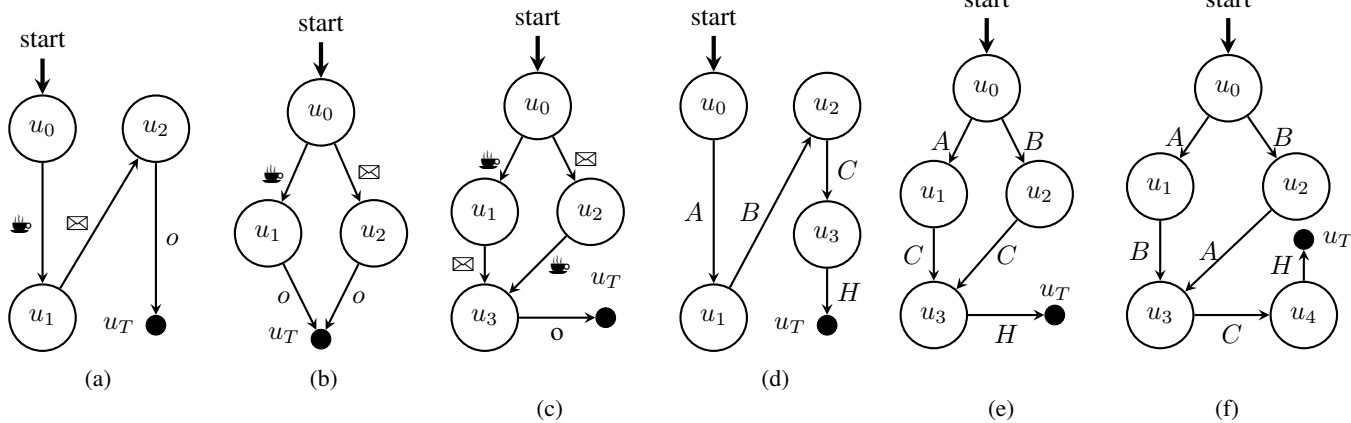

Figure 4: Finite state automatons for the Office domain (sequential (a), disjunction (b) and composite (c)) tasks and the Delivery domain (sequential (d), disjunction (e) and composite (f)) tasks.

However, the majority of the methods in this area are designed for single-task solutions, with only several focusing on acquiring a set of policies that is capable of addressing multiple tasks (Toro Icarte et al. 2018b; León, Shanahan, and Belardinelli 2020; Kuo, Katz, and Barbu 2020; Araki et al. 2021; Vaezipoor et al. 2021).

From these works, our approach is the most similar to the Logical Options Framework (Araki et al. 2021). The main difference is that LOF trains a single policy for each sub-goal, resulting in a set of learned policies that is either smaller than or equal to the set acquired through our method. While employing one policy per sub-goal proves sufficient for obtaining a globally optimal policy through planning in deterministic environments (Wen et al. 2020), this may not hold true in stochastic environments, as our experiments demonstrate. In such instances, the policies generated by LOF are hierarchically optimal but fall short of global optimality.

Two notable examples from aforementioned works on multi-task learning with formal language specifications are the works of Toro Icarte et al. (2018b) and Vaezipoor et al. (2021). The former struggles with generalizing to unseen tasks, because it uses LTL progression to determine which sub-tasks need to be learned to solve given tasks. The Q-functions that are subsequently learned for each LTL sub-task will therefore not be useful for a new task if its sub-tasks were not part of the training set. Such limitation does not apply to the latter as it instead encodes the remaining LTL task specification using a neural network and conditions the policy on this LTL embedding. While this approach may be more adaptable to tasks with numerous propositions or sub-goals, it risks generating sub-optimal policies as it relies solely on the neural network to select the next proposition to achieve, without incorporating planning. Additionally, since the planning is implicitly done by the neural network, the policy is less interpretable than when explicit planning is used.

The method we propose can be viewed as a method for composing value functions through successor features, akin to previously proposed approaches for composition of value functions and policies (van Niekerk et al. 2019; Barreto

et al. 2019; Nangue Tasse, James, and Rosman 2020; Infante, Jonsson, and Gómez 2022). In the work of Infante, Jonsson, and Gómez (2022), which is the closest to our work, the authors propose to learn a basis of value functions that can be combined to form an optimal policy. However, unlike our method, their approach only works in a restricted class of linearly-solvable MDPs. Lastly, since our approach uses the values of exit states for planning it is also related to planning with exit profiles (Wen et al. 2020). The CCS that we propose to use as a policy basis in our work can be seen as a collection of policies that are optimal for all possible exit profiles.

## Discussion and Conclusion

In this work, we address the problem of finding optimal behavior for new non-Markovian goal specifications in known environments. To do so, we introduce a novel approach that uses successor features to learn a policy basis, that can subsequently be used to solve any unseen task specified by an FSA with the set of given predicates $\mathcal{P}$ by planning. Our method is the first that can provably generalize to such new task specification without sacrificing optimality in both deterministic and stochastic environments.

The experiments show that our method offers several advantages over previous methods. First, due to the use of SF, it allows for faster composition of the high-level value function since it drastically reduces the number of states to plan on. Secondly, thanks to using a CCS over a set of options our method achieves optimality even in stochastic environments (as shown in the Double Slit example). Lastly, we do not require that there exists a single exit state per predicate which permits more flexible task specification while at the same time allowing deployment in more complex environments.

A limitation of our approach could be the need to construct a full CCS if one wants to attain global optimality. While the construction of CCS is not timecomsuming for environments with several exit states presented in our work, the computation cost of finding the full CCS could become too large for environments with many exit states. In such case one could instead learn a partial CCS at the cost of a bounded

decrease in performance (Alegre, Bazzan, and Da Silva 2022) or consider splitting the environment into smaller parts with less exit states. While our experiments only considered discrete environments, our method should also be applicable in continuous environments with minor adjustments. These include: using an contiguous set of states instead of a single exit state and using reward shaping to facilitate learning in sparse reward setting.

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
