# OpenReview forum: "Planning with a Learned Policy Basis to Optimally Solve Complex Tasks"
_icaps-conference.org/ICAPS/2024/Conference — ICAPS 2024_

### Official Review · Reviewer_bUJj · 2023-12-23

**Significance And Importance:** 3
**Soundness:** 4
**Novelty:** 3
**Clarity:** 3
**Overall Evaluation:** 2
**Confidence:** 3

**Weaknesses:**

2: No major or minor weaknesses.

**Contributions Of The Paper:**

The paper proposes a novel methodology to learn sub-task policy approximations, with optimality guarantees on the full task. One important aspect is that learned sub-policies can be composed with higher versatiliy, achieving faster convergence to full task optimum. This holds true also in non-deterministic scenarios, as the double split, where better performance is achieved with respect to established competitors.

**Ethical Considerations:**

(5) Excellent: The paper comprehensively addresses all of the applicable ethical considerations

**Nomination For Best Paper:**

Yes

**Questions For Authors:**

I just require to fix minor presentation weaknesses as above.
Also, it would be useful for the community to have access to an open source implementation of the methodology. I encourage the authors to share this, upon acceptance.

**Reproducibility:**

2: Some details are missing, but the paper still appears to be replicable with some effort.

**Strengths Of The Paper:**

The paper gives an important contribution, appears technically and theoretically sound and the experimental validation is very convincing. The state of the art is well discussed.

**Weaknesses Of The Paper:**

I just suggest the authors to include a leading example in the methodological section, e.g., from the office domain, for easier readability.
Also, please clarify in the caption the meaning of lines in figure 1a (office domain sketch).

---

> ### Author Rebuttal · Authors · 2024-01-26
>
> We thank the reviewer for their feedback. We will disclose a link to the code in the final version of the paper.

---

### Official Review · Reviewer_Bytc · 2024-01-11

**Significance And Importance:** 2
**Soundness:** 2
**Novelty:** 2
**Clarity:** 2
**Overall Evaluation:** 1
**Confidence:** 3

**Weaknesses:**

0: Minor weaknesses requiring some work to be addressed for the paper to be accepted.

**Contributions Of The Paper:**

This work proposes a method to learn a set of local policies in sub-tasks such that their combination forms a globally optimal policy for different problems described with FSAs.
The contributions are the following:
(1)  learning approach that is suitable for planning in stochastic domains based on successor features.
(2) a planning framework that uses such a policy
(3) a demonstration that the framework produces optimal solutions.

**Ethical Considerations:**

(1) Not Applicable: The paper does not have any ethical considerations to address

**Nomination For Best Paper:**

No

**Questions For Authors:**

The questions are reported below:

(1) Please explain the domain in E.g., in the office domain depicted in Figure 1a with a sentence under the “Propositional Logic” section. This helps to clarify the logical consequence.

(2) The sentence “This inherently makes obtaining the optimal policy more efficient than solving the
whole product MDP, as we reduce the number of states on which it is necessary to compute the value function” is very strong. It has to be demonstrated.

(3) The sentences “we can define an arbitrary vector w of rewards on the exit states, and use the CCS to obtain an optimal policy π  and an optimal value function V w without learning. We can then use composition by setting the reward of the exit states equal to the optimal value” needs to be clarified. How the optimal value is computed?

(4) The computation of Equation 11 and the following steps for achieving that it converges to optimal weights should be further explained.

(5) In the experiment section, the ways to train the sub-subpolicies should be mentioned (which algorithms, which rewards, number of epochs, number of steps per epoch, hardware for training, etc.)

(6) Similarly to the previous point, please specify the parameters related to the next re-training.

(7) “We use a total of 8 · 10 3 samples per subpolicy in both environments” → Please check the number of samples with respect to the results in Figure 2.

(8) Additionally, since the planning is implicitly done by the neural network, the policy is less interpretable than when explicit planning is used. → Explain and motivate how the proposed approach is more explainable.

(9) Given the discussion reported at the end of the paper, it would be interesting to compare the proposed approach with respect to the baseline in terms of computation cost.


Minor:

(10) Please specify the meaning of FSA (finite state automaton (FSA)) also in the introduction not only in the abstract

(11) The figure should be cited in ascendant order in the text. Fig. 4 appears sequentially after Figure 1 and before Figure 1.

(12) Typo: provably

**Reproducibility:**

1: Difficult to reproduce because of missing detail.

**Strengths Of The Paper:**

The strength of the paper is its completeness. This paper includes both theoretical formulation of the problem and experiments including a comparison of the performance of the proposed approach with the state-of-the-art baseline.

**Weaknesses Of The Paper:**

This paper presents a theoretical description of the problem that is not completely clear on some points. For instance:
1) Authors mentioned that In line with the previous reasoning, we introduce a two-step algorithm in which the agent first learns a Π CCS (a set of policies that constitute a CCS) on a well-specified representation of the environment.” → However, it is not clear the difference between the proposed approach with respect to the state-of-the-art methods that pre-train the policies that are tuned for specific tasks.
(2) Please see the other aspects mentioned in the next section.

Another weakness of the paper is the lack of a link/detailed description of the planning framework that uses the proposed policy (such a framework is mentioned among the contributions of the paper). It would be helpful to add the link to the code.

---

> ### Author Rebuttal · Authors · 2024-01-26
>
> Our approach consists of (a) building a policy basis (CCS) and (b) using a VI-like algorithm to obtain the overall solution by combining the policies obtained in (a). Our intention is never to fine-tune policies in (a) , but rather achieve ‘zero-shot’ shot learning for the overall solution of the FSA with the planning algorithm in (b).
>
> (2) When solving the whole (flat) product MDP MxF, comparable algorithms have to obtain a value for  O(|U|·|S|) states. Our algorithm, due to the compositionality of the SF framework, simply has to iterate over the O(|U|·|E|) set of exit states instead (Algorithm 1 and proof reflect this). See reply to **Reviewer SsV3 (Minor points, 1)** for more details.
> (3) The optimal value is a consequence of Equation [8] (and Theorem 1).  Assuming a complete CCS, for any arbitrary w, V*(s, u) = max_a Q*(u, s, a).
>
> (4) The contraction mapping theorem states that a sequence of linear updates converges to a unique fixed point if the multiplier in each update is strictly less than 1, which we prove is the case. We agree that this could be made more explicit in the proof, and we will do so in future versions.
>
> As for (5), (6) and (7): we do not consider sub-subpolices. Except for the hardware, the details for the training of subpolicies (policies in the CCS) are described in the manuscript (rewards, num of samples and algorithms). Our planning approach (in Algorithm 1) is a value iteration-like algorithm with no parameters. This step simply uses the subpolicies in the CCS to retrieve the global solution. After a double check, the numbers are indeed correct and consistent with what the figures show.
> (8) We consider the solution produced by our method to be more interpretable because it relies on a VI-like algorithm for planning. Though neither the CSS nor the policies in it might be interpretable themselves, the overall policy is more interpretable than a NN. Please see our reply to **Reviewer SsV3 (Minor points, 2)** for more details.
>
> (9) We tried to state our case in the manuscript (lines 424-432). We insist on the reply to (2). Our algorithm needs to compute the value function in a smaller set of states than previous approaches. When the size CCS is small, then each iteration of our algorithm is cheaper. Besides, it needs less iterations to converge to the optimal solution (Figure 2, right column). In the final version of the paper we could append runtime experiments.
>
> We consider (1,10,11,12) for the final version of the manuscript.

---

### Official Review · Reviewer_SsV3 · 2024-01-22

**Significance And Importance:** 2
**Soundness:** 3
**Novelty:** 2
**Clarity:** 3
**Confidence:** 3

**Weaknesses:**

0: Minor weaknesses requiring some work to be addressed for the paper to be accepted.

**Contributions Of The Paper:**

-------------- Post-Response ---------------

Thank you for your response. Given that the CCS is a very new technique, I think more details about it should be included in this work to make it more self-contained. While I appreciate there are space constraints, at least some more detail should be included due to the centrality to that method to your work.
------------------------------------------------

The main goal of the paper is to do transfer learning on tasks with a well-defined low-level environment and a high-level propositional labelling that can be used to define non-Markovian tasks. The propositions are then used as to define the successor representation of the MDP. Intuitively, this creates a set of Markovian tasks, one for achieving each proposition (roughly speaking). This is then used to identify a set of policies, the Convex Coverage Set (CCS), which collectively can be combined for optimally achieving any Markovian reward given by the linear combination of the achieving features.

The CCS generation approach was previously known, but this paper identifies how the CCS can then be used to generate optimal policies for a non-Markovian reward expressed as a finite-state automata (FSA) over the propositions. Specifically, the method performs a value iteration-like process that learns a per-FSA state weight vector. This method converges to the optimal solution for the new task without any learning.

The method is experimentally compared against the Logical Options Framework, and a basic Q-learning baseline, on two small domains. The new approach converges faster to the optimal solutions in all cases.

**Ethical Considerations:**

(1) Not Applicable: The paper does not have any ethical considerations to address

**Nomination For Best Paper:**

No

**Overall Evaluation:**

-1: (weak reject)

**Questions For Authors:**

1) Please clarify how the charts correspond to a planning phase versus a learning phase. Perhaps a summary of how the CCS and the successor feature vectors for these is computed would be helpful here.
2) Please clarify how the method would scale with a larger set of exit states vs the number of propositions.

**Reproducibility:**

3: Authors describe the implementation and domains in sufficient detail.

**Strengths Of The Paper:**

The paper brings a new and very interesting tool in the CCS and its use for transfer learning to this community. It also makes the clever and interesting step forward by showing how it can be used to handle non-Markovian rewards, which has been of great interest recently.

The theoretical results are powerful here in that they show something quite strong about the use of a CCS in non-Markovian tasks.

I have some issues with the experiments as I detail in the next section, but they do show potential in this method. In did appreciate the Double slit experiment as an explanation of the issues with LOF.

**Weaknesses Of The Paper:**

I found the abstract/introduction initially hard to reconcile with the rest of the paper. The abstract and introduction talk about local policies, and I had difficulty seeing the connection with the rest of the paper. I eventually came to understand these as the policies in the CCS which one would expect to attempt to reach the terminal states. But this connection was left largely implicit for the rest of the paper, and I think this should be better connected.

I also am unclear about some of the limitations of this method. Constructing the CCS is listed as one, but it also seems to rely on the fact that the set of tasks that are possible can only be captured by an (arbitrarily complex) non-Markovian task that involves visiting a small number of exit states. In particular, since a successor feature is created per exit state (instead of per proposition), it seems like that must be kept low to make this feasible. However, I would be interested in hearing more comment on this topic.

While I suppose this is left to future work, there is also the big question of how this would extend to a function approximation setting.

I had some issues understanding the experimental design, partially because I am not very familiar with either CCS or LOF. In both cases, it feels like a bit more background is needed to understand the described method and how they were evaluated. In the case of CCS, I think further information is needed on the how the policies are generated. The results section suggests these are generated sequentially, but are elements of that set added to or discarded? Or is it some sort of process where one is added at a time until some sort of "complete set" is found. Or are the policies generated first, then evaluated? The new method also seems like it would involve the initial learning phase, following by the planning phase after which learning is no longer needed. Its unclear to me in the charts where this crossover happens.

Because of the distinct learning vs planning phase, I am unsure how to think about runtime here, especially in comparison to LOF (which I assume is a standard non-planning method?). How long does the planning take, for example?

In general, the experiments do not feel especially comprehensive, as they focus on two relatively small problems, and a small amount of tasks between them. They do show what you would hope for, but they do not go in especially great detail.

Minor Points
- Related to this point about learning vs planning time, it might be helpful to see how long it takes to converge to the optimal policy on a single task after the CCS has been computed. How this compares to LOF for the different tasks could be interesting
- Is the CCS really explainable? It finds 12 policies for Office World and 15 for Delivery. Those are non-intuitive numbers, and I'm curious how one could extract meaning from the found policies.

---

> ### Author Rebuttal · Authors · 2024-01-26
>
> We have indeed used several terms: ‘local policies/subpolicies/base policies/policy basis’ to refer to the same concept. We agree that by clarifying these and making them consistent throughout the paper would make the paper easier to understand. We will make this change in the revision/final version of the manuscript .
>
> Our method should be directly applicable to discrete domains with func. approximation (FA) settings since successor features have been shown to work in such settings.The main challenge in continuous state spaces is that exit states need to be adjusted to small contiguous regions of states. If these regions are small and have approximately the same value function the planning should not be compromised. Similarly to any FA setting, the global optimality guarantees may no longer be preserved.
>
> **To the questions**
>
> (1) The CCS is indeed built sequentially, adding elements to such a set until a complete set is found. For further details of this process we refer the reviewer to Algorithm 1 in Alegre et al. 2022. Both learning phases are slightly different (CCS learns its elements sequentially, LOF pre-defines the options and learns them alternately), but the planning step is completely comparable. Since the learning phases (and the number of learned policies) differ, we show the performance of the overall solution to the FSA as the CCS policies (resp. LOF options) are learned in Figure 2 (left column).
>
> (2) The size of the CCS does not depend on the number of propositions, but it does depend on the number of exit states. However, at this point, we cannot express such a dependency with a formal expression.
>
> **Minor points**
>
> (1) This is what Figure 2 (right column) shows, though averaged for 3 tasks. Here we assume that CCS (resp. optimal options) are given at the beginning. Now we compare the number of planning iterations of LOF (see their paper, Algorithm 1) vs. our Algorithm 1. Iterations are of the cost of O(|U|·|S|·|E|) and O(|U|·|E|·|CCS|) respectively. Thus (1 iteration of) our algorithm is more efficient when the size of the CSS is smaller than the number of states of the MDP.
>
> (2) We point out that we do not claim that CCS (or its policies) are explainable. However, we argue that the VI planning is a very transparent process that makes it clear why a certain sub-policy was chosen and how the overall solution is composed. This is much more intuitive than if the planning is done implicitly in a (black box) NN (as described in lines 58-68)

---

### Meta-Review · Area_Chair_4LrE · 2024-02-06

**Recommendation:** Accept (Oral)
**Confidence:** 4

**Metareview:**

The basic idea of the paper is a technique for learning local policies for sub-tasks that can be composed to obtain optimal global policies for non-markovian composite tasks where the reward is a linear combination of the sub-tasks.

The reviewers agree the method is interesting and there is novelty in the idea.

Two major points were discussed by the reviewers.

1. Clarity: The paper is quite dense in the technical explanations and the CCS should be better reported in the main text to make the paper self-contained. In general, a high-level description of the approach at the beginning would help the understanding of the whole article.
2. The experiments are not very comprehensive though, but they do show a reasonable amount of potential.

All in all, the general agreement was that the paper has no major flaws and introduces interesting ideas, even if there is room for improvement especially in experiments and presentation. We recommend the authors to report the rebuttal answers in the main text and make any effort to improve the clarity of the presentation.

**Ethical Considerations:**

(1) Not Applicable: The paper does not have any ethical considerations to address